# A Comprehensive Profiling of Cellular Sphingolipids in Mammalian Endothelial and Microglial Cells Cultured in Normal and High-Glucose Conditions

**DOI:** 10.3390/cells11193082

**Published:** 2022-09-30

**Authors:** Koushik Mondal, Richard C. Grambergs, Rajashekhar Gangaraju, Nawajes Mandal

**Affiliations:** 1Department of Ophthalmology, Hamilton Eye Institute, The University of Tennessee Health Science Center, Memphis, TN 38163, USA; 2Department of Anatomy and Neurobiology, The University of Tennessee Health Science Center, Memphis, TN 38163, USA; 3Department of Pharmaceutical Sciences, College of Pharmacy, The University of Tennessee Health Science Center, Memphis, TN 38163, USA; 4Memphis VA Medical Center, Memphis, TN 38104, USA

**Keywords:** sphingolipids, Hexosyl-ceramide, Lactosyl-ceramide, Sphingosine-1-phosphate, high-glucose, endothelial cells, microglial cells, inflammatory markers, proliferative markers

## Abstract

Sphingolipids (SPLs) play a diverse role in maintaining cellular homeostasis. Dysregulated SPL metabolism is associated with pathological changes in stressed and diseased cells. This study investigates differences in SPL metabolism between cultured human primary retinal endothelial (HREC) and murine microglial cells (BV2) in normal conditions (normal glucose, NG, 5 mM) and under high-glucose (HG, 25 mM)-induced stress by sphingolipidomics, immunohistochemistry, biochemical, and molecular assays. Measurable differences were observed in SPL profiles between HREC and BV2 cells. High-glucose treatment caused a >2.5-fold increase in the levels of Lactosyl-ceramide (LacCer) in HREC, but in BV2 cells, it induced Hexosyl-Ceramides (HexCer) by threefold and a significant increase in Sphingosine-1-phosphate (S1P) compared to NG. Altered SPL profiles coincided with changes in transcript levels of inflammatory and vascular permeability mediators in HREC and inflammatory mediators in BV2 cells. Differences in SPL profiles and differential responses to HG stress between endothelial and microglial cells suggest that SPL metabolism and signaling differ in mammalian cell types and, therefore, their pathological association with those cell types.

## 1. Introduction

Sphingolipids (SPLs) are a diverse and ubiquitous family of lipids that function as structural components of the cellular membranes and have signaling roles in various cellular processes, including cell growth, cell adhesion, inflammation, and apoptosis [1,2,3,4]. Biochemically, all SPLs contain an amino-alcohol backbone and a covalently bound fatty acid forming the base structure of an SPL, the Ceramide (Cer). Cer can have numerous possible head groups and a variety of fatty acid side chains, allowing for a large array of unique species with diverse bioactivity. SPL metabolism is complex, and SPL subspecies may have unique roles in different cell types [5]. Besides being integral structural components of the biomembrane, SPL species play signaling roles. The key SPL metabolite, Cer, signals for inflammation and apoptosis. However, Cer-derived SPL, Sphingosine-1-phosphate (S1P), regulates the processes such as cell survival, proliferation, and formation of cellular junctions [2]. Thus, by altering and maintaining the balance between various bioactive species, cellular SPLs may play a major role in maintaining cellular homeostasis under normal conditions and under stresses [6,7,8]. Further, their involvement in human diseases showcases the importance of SPL metabolism and signaling in human physiology as SPL metabolic derangements are associated with several diseases, including Farber, Tay–Sachs/Sandhoff, Gaucher, Krabbe, and Niemann Pick disease [9,10].

Besides metabolic disorders, studies in the past two decades revealed SPL’s and bioactive SPL’s association with various human diseases, including neurodegenerative, inflammatory, neovascular, neoplastic, and diabetes mellitus (DM) [9,11]. Abnormal lipid metabolism (dyslipidemia) is a recognized pathophysiological factor for microvascular and macrovascular complications in DM. Alterations of SPL metabolism are also gaining recognition in the pathophysiology of DM and many other chronic inflammatory disorders [11,12,13]. SPL metabolic changes are suspected to be one of the root causes of developing diabetic complications such as diabetic retinopathy (DR) and diabetic neuropathy (DN) [14,15]. Endothelial cells maintain vascular permeability by forming a continuous monolayer covering the inner lumen of blood vessels and help regulate solute trafficking, immune responses, and angiogenesis [16]. Breakdown or leakage of the endothelial barrier is one of the pathological hallmarks of diabetic microvascular complications, as occurs in cases of DR [17]. Studies report that dysfunctional retinal vasculature and compromised vascular repair process in DM are associated with activation of a major Cer-generating enzyme, acid sphingomyelinase (aSMase), in retinal endothelial cells (HREC) [18,19]. Microglial cells, on the other hand, are resident macrophages in the central nervous system (CNS) and respond to infection and tissue injury as the primary effectors of CNS inflammatory responses [20]. These immune cells are believed to be a primary driver of neuroinflammation-related neurodegeneration [21]. Increased levels of Cer via activation of SMase in microglia activate proinflammatory factors [22]. Exposure to high-glucose conditions has been shown to stimulate proinflammatory cytokine signaling in microglial cells [23,24]. Microglia can contribute to perivascular inflammation in diseases such as DR, which exacerbates vascular pathology and neovascularization [25].

While SPL signaling is implicated in critical physiological processes governed by endothelial and microglial cells, there is still little information regarding the composition of SPLs within these distinct cell types and how they modulate their SPL profile when exposed to higher environmental glucose, such as in the cases of DM. The present study investigated the SPL profiles of human retinal endothelial cells (HREC) and murine-derived microglia (BV2 cells) in normal physiologic conditions and in high-glucose conditions and the expression and function of their metabolic enzymes and inflammatory and vascular permeability markers. Significant differences were observed in SPL profiles between these cell types at baseline and in response to high-glucose stress. Along with the changes in their profiles, we also observed changes in the expression and function of their metabolic enzymes, vascular permeability factors, and proinflammatory markers in both cell types in response to high-glucose stress. These findings could reflect innate differences in SPL metabolism and SPL mediation of cell processes in response to cellular stress between endothelial and microglial cells.

## 2. Materials and Methods

### 2.1. Culturing of Human Retinal Endothelial Cell and Murine Microglial Cell

Human retinal endothelial cells (HREC) (ACBRI 181) were obtained from Cell Systems, Kirkland, WA, USA. They were cultured in 75 cm^2^ culture flasks with Cell Systems Complete Medium (4NO-500) (Cell Systems, Kirkland, WA, USA) and Culture Boost (4CB-500) (Cell Systems, Kirkland, WA, USA), following the manufacturer guidelines. The cultures were maintained at 37 °C in a humidified atmosphere containing 95% air and 5% CO_2_. Before plating the cells, the plates were treated with Attachment Factor (4ZO-201) (Cell Systems, Kirkland, WA, USA). The cells were then plated in 6-well plates with 1 × 10^6^ cells/well (passage 6–7) following the same procedure. After 24 h of incubation, media were replaced with fresh media containing 5 mM glucose for control as normal glucose (NG) with 20 mM of glucose (NG + 20 mM D-glucose) as high glucose (HG) and cultured for 48 h to represent diabetic endothelial cells, as reported earlier [26]. In addition to normal glucose (5 mM), cells were also cultured with 20 mM of L-glucose (LG; NG + 20 mM L-glucose) or 20 mM of mannitol (Man; NG + 20 mM mannitol) served as additional controls. Subsequently, cells were harvested for sphingolipidomic and biochemical analyses. The mouse microglial cell line, BV2, was a kind gift from Professor Grace Sun, Ph.D., University of Missouri, Columbia, MO, USA. BV2 cells were routinely grown in 100 mm cell culture dishes in Dulbecco’s Modified Eagle Medium (DMEM) with 10% fetal bovine serum and antibiotics, as described previously [27]. BV2 cells were exposed to normal or high-glucose conditions described above and proceeded for sphingolipidomic, molecular, and biochemical analyses.

### 2.2. Mass Spectrometry Analysis of Sphingolipids

Following previously published procedures, sphingolipids in HREC and BV2 were quantified and analyzed in the Lipidomic Core facility at Virginia Commonwealth University, Richmond, VA, USA [7,28]. Internal standards were procured from Avanti Polar Lipids (Alabaster, AL, USA) and added to samples at a ratio of ethanol/methanol/water (7/2/1) as a cocktail of 500 pmol each. Standards for sphingoid bases and sphingoid base 1-phosphates were 17-carbon chain length analogs: C17 Sphingosine, C17 Sphinganine, C17 Sphingosine-1-Phosphate, and C17 Sphinganine-1-Phosphate. Standards for N-acyl sphingolipids were C12-fatty acid analogs: C12 Sphingomyelin, C12 Ceramide, C12 Glycosylceramide, C12 Lactosylceramide, C12 Ceramide-1-Phosphate. For LC-MS/MS analyses, a Shimadzu LC-20 AD binary pump system coupled to an SIL-20AC autoinjector and DGU20A3 degasser coupled to an ABI 4000 quadrupole/linear ion trap (QTrap) (Applied Biosystems, Foster City, CA) operating in a triple quadrupole mode was used. Q1 and Q3 were set to pass molecularly distinctive precursor and product ions (or a scan across multiple *m*/*z* in Q1 or Q3), using N2 to collisionally induce dissociations in Q2 (which was offset from Q1 by 30–120 eV); the ion source temperature set to 500 °C.

Samples were collected in borosilicate tubes (13 × 100 mm), and 1 mL of CH_3_OH and 0.5 mL of CHCl_3_ were added along with a cocktail of internal standards. Then, samples were dispersed with ultra sonicator for 30 s at room temperature and incubated overnight at 48 °C. After cooling the mixture, 75 µL of 1 M KOH in CH_3_OH was added. To cleave potentially interfering glycerolipids, the mixture was then sonicated briefly and incubated in a shaking water bath for 2 h at 37 °C. Using 6 µL of glacial acetic acid, the extract was brought into neutral pH, then centrifuged, and the supernatant was collected and transferred into a new tube. The extract was dried using a Speed Vac. The dried residue was reconstituted in 0.5 mL of starting mobile phase solvent for LC/MS and sonicated for 15 s, preceded by centrifugation. The supernatant was collected and transferred to an autoinjector for analysis.

A combination of C18 and LC-NH2 columns were used to analyze all species of sphingolipids following the methodology developed by Dr. Alfred Merill [29]. To separate complex sphingolipids and sphingoid base 1-phosphates, reverse phase LC using a Supelco 2.1 (internal diameter) ×50 mm Ascentis Express C18 column (Sigma, St. Louis, MO, USA) and a binary solvent system at a flow rate of 0.5 mL/min with column oven set at 35 °C was used. Before injection of the sample, the column was equilibrated with a solvent mixture of 95% mobile phase A1 (CH_3_OH/H_2_O/HCOOH, 58/41/1, *v*/*v*/*v*, with 5 mM ammonium formate) and 5% mobile phase B1 (CH_3_OH/HCOOH, 99/1, *v*/*v*, with 5 mM ammonium formate) for 5 min, and after sample injection (typically 40 µL), the A1/B1 ratio was maintained at 95/5 for 2.25 min, followed by a linear gradient to 100% B1 over 1.5 min, which was held at 100% B1 for 5.5 min, followed by a 0.5 min gradient return to 95/5 A1/B1. The column was re-equilibrated with 95:5 A1/B1 for 30 s before the next run. The species of Cer, HexCer, SM, and sphingoid lipids such as sphingosine (Sph), dihydro-sphingosine (Dh-Sph), S1P, and Dh-S1P were identified based on their retention time and *m*/*z* ratio and quantified as described in previous publications [28,29,30,31].

### 2.3. Quantitative RT-PCR

Total RNA from HREC and BV2 cell pellets was extracted using the Ambion RNA mini extraction kit following the manufacturer’s protocol (Ambion TRIzol^®^ Plus RNA purification kit: Life Technologies, Carlsbad, CA, USA); cDNA synthesis was carried out by SuperScript™ IV First-Strand Synthesis SuperMix (Invitrogen, Carlsbad, CA, USA). Quantitative RT-PCR was performed according to our previously published procedure [4,7]. The sequence of the primers for human gene expression (for HREC) and rodent gene expression (for BV2) are provided in Appendix A.

### 2.4. Immunocytochemistry

For immunocytochemical analysis, cells were seeded on glass coverslips and cultured overnight. The following day, cells were incubated for 3, 24, and 48 h in normal and high-glucose conditions, washed, and fixed with 4% paraformaldehyde for 20 min at room temperature. Cells were incubated with anti-LacCer antibody (CD17, Santa Cruz, SC 65253), followed by secondary antibody (Alexa Fluor 568; Invitrogen catalog # A11037), and mounted with ProLong Diamond Antifade mountant with DAPI (Invitrogen, Carlsbad, CA, USA), then examined using a Zeiss 710 confocal laser scanning microscope (Carl Zeiss, Thornwood, New York, NY, USA) Images were captured using ZEN 2012 imaging software. The images were analyzed and quantified by ImageJ software from 5–6 field/image per condition and from at least three independent experiments.

### 2.5. Sphingomyelinase Assays

The enzymatic activities of acidic and neutral sphingomyelinase (aSMase and nSMase, respectively) were measured in HREC and BV2 using the Amplex Red Sphingomyelinase Assay Kit (Invitrogen, Carlsbad, CA, USA) following previously published protocols [4,32].

### 2.6. Data Analysis

Statistical analysis was performed using GraphPad Prism 8 (GraphPad Software, San Diego, CA, USA), and one-way ANOVA (for more than 2 groups) or *t*-test (between 2 groups) were used to calculate the significant difference between the groups. Data are presented as mean ± SEM. Statistical significance was carried out on raw data and *p* < 0.05 was considered as significant.

## 3. Results

### 3.1. Comprehensive Sphingolipid Profiles of Human Retinal Endothelial Cells and Murine Microglial Cells

Bioactive SPL signaling is involved in numerous cellular processes in various cell types. The present study characterized the baseline sphingolipid profiles of two cell lines with distinct biological roles, HREC and BV2 cells. No report was found on comprehensive baseline SPL profiles of retinal endothelial cells and microglial cells. We cultured HREC and BV2 cells with media containing 5 mM glucose for 48 h, then performed sphingolipid analysis by LC-MS/MS. HREC were found to have total SPL levels of 220 pmol/mg protein, while the relative composition of major SPL classes was 79% SM, 14% Cer, 4% LacCer, and 3% HexCer (Figure 1A,C). BV2 cells, on the other hand, were found to have total SPL levels amounting to 130 pmol/mg protein, and the relative composition of major SPL classes was 78% SM, 6% Cer, 13% HexCer, and 3% LacCer (Figure 1B,D).

While both the cell lines have similar levels of SM, they significantly differ in the composition of Cer and HexCer. Further analysis revealed differences in the mole percentage of various SPL species (Cer, SM, HexCer, and LacCer) between HREC and BV2 cells (Figure 2). In HREC, short-chain species of Cer (C16:0 and C18:0) were found to be higher compared to BV2 cells (Figure 2A,B), whereas long-chain Cer species (C24:0 and C24:1) were higher in BV2 cells (Figure 2A,B). Similarly, long-chain SM levels were higher in BV2 cells, as well as C14:0 and C22:0 species (Figure 2C,D), whereas, C16:0 SM level was higher in HREC (Figure 2C,D). HexCer composition was dominated by C22:0 and C24:0 species, which accounted for 70% of HexCer in HREC (Figure 2E), while about 50% of HexCer was distributed between carbon chains C22:0 (16%) and C24:0 (35%), and the remaining 50% were distributed between C16:0 (26%) and C24:1 (21%) in BV2 (Figure 2F). Differences were also noted in short-chain LacCer composition between HREC and BV2 (C16:0: 23% vs. 20%; C18:0: 6% vs. 1%; C22:0: 10% vs. 6%) (Figure 2G,H). In BV2, 73% of LacCer species composition were long-chain, mainly C24:0 and C24:1, while in HREC, they accounted for approximately 60% of LacCer (Figure 2G,H).

### 3.2. High-Glucose-Induced Changes in Sphingolipid Profile in Cultured HREC and BV2

After determining the composition of different SPL species in endothelial and microglial cells under normal conditions, we investigated the SPL profiles in HREC and BV2 cells exposed to high-glucose conditions (HG; NG + 20 mM D-glucose) for 48 h. HREC and BV2 cells cultured in normal glucose (5 mM D-glucose), high L-glucose (LG; NG + 20mM L-glucose), or mannitol (Man; NG + 20mM mannitol) served as osmolarity controls. HG treatment in HREC cells showed a twofold increase in LacCer levels and a significantly reduced level of SM relative to NG HREC (Figure 3A). The relative percent composition of LacCer in HG-treated cells was 13% (* *p* < 0.05) (Figure 3B(iv)), whereas it was within 5% to 7% in control conditions (NG, LG, and Man) (Figure 3B(i–iii)). Among the individual LacCer species, significant increases were found in C16:0 (** *p* < 0.01); C22:0, C24:0, and C24:1 (* *p* < 0.05) in HG (Appendix A). The relative composition of SM in HG-cultured cells was 69% (Figure 3B(iv)) compared to 79%, 77%, and 78% in NG, LG, and Man, respectively (Figure 3B(i–iii)), indicating a decrease in relative SM composition under HG. No changes were observed in minor bioactive long-chain base (LCB) SPL metabolites, such as Sph, dhSph, and S1P in HG-cultured HREC cells, but there was a significant decrease in dhS1P levels (* *p* < 0.05) (Appendix A). The immunocytochemical analysis further revealed accumulation of LacCer in HG-treated HREC at all time points investigated, with data reaching statistical significance at 48 h compared to NG cells (* *p* < 0.05, Figure 4).

HG treatment in BV2 cells revealed a significant increase in the level of total HexCer (40.0 vs. 17.0 pmol/mg protein; ** *p* < 0.01) (Figure 5A). In addition, the relative composition of HexCer increased by approximately 2–3-fold (Figure 5B(iv)) compared to NG, LG, and Man (Figure 5B(i–iii)). Analysis of individual HexCer species showed significant increases in C14:0 (*** *p* < 0.001), C18:1, C18:0 (** *p* < 0.01), C22:0, C24:0, and C24:1 (* *p* < 0.05) relative to NG (Appendix A). Increases in HexCer coincided with significant decreases in the levels of SM and LacCer in HG-treated BV2 compared to NG (* *p* < 0.05). The relative composition of SM in HG-treated BV2 was reduced to 50% of total SPLs compared to NG (77%), LG (73%), and Man-cultured cells (69%) (Figure 5B). HG treatment also reduced the levels of Sph and dhS1P, and there was a significant increase in S1P levels (*** *p* < 0.001) (Appendix A).

Further comparison of the bioactive SPLs and their relative compositions indicated a significant increase in the ratio of Cer:Sph (Figure 6C) in HG-treated HREC, suggesting an accumulation of Cer. However, HG in BV2 induced a significant increase in the ratios of Cer:SM and S1P:Sph, suggesting an accumulation of S1P (Figure 6B,F). The 2.5-fold increase in S1P:Sph ratio in HG-treated BV2 was not reflected in HG-treated HREC (Figure 6E,F).

### 3.3. High-Glucose-Induced Changes in Very-Long-Chain Sphingolipid Composition in HREC and BV2

VLC SPLs, a special group of SPLs, have recently been under active investigation for their recognized roles in various human diseases and their association with the function of the vascular endothelial cells [33]. In HREC cells, HG treatment caused a significant decrease in VLC Cer and VLC SM species of carbon chain lengths C28 to C30, and a significant increase in C26:1 and C26:0 VLC SM (Figure 7A,B). A similar trend in VLC LacCer was noted with an approximately 2.5-fold increase in C26:1 and C26:0 VLC LacCer, but there were significant decreases in C28:1 and C28:0 species in HG-treated HREC (Figure 7D). The elevations of C26:1 and C26:0 fatty acids of Lac-Cer and SM, along with concomitant decreases in C28 and C30 fatty acids in all SPL, suggest that the function of the enzyme, elongation of very-long-chain fatty acid-like 4 (ELOVL4) that converts C26 fatty acids to higher-chain-length fatty acids, might have been affected in HG. In line with this observation, when gene expression was measured, *ELOVL4* expression was found to be significantly lower in HG-treated HREC than NG (Figure 7E). *ELOVL4* expression was reduced in control cells at after 48 h compared to 3 h (* *p* < 0.05). *ELOVL4* expression was significantly reduced by HG at three hours (* *p* < 0.05) compared to controls (Figure 7E) and was further suppressed at 48 h (* *p* < 0.05) (Figure 7E).

In BV2 cells, a significant increase in C26 to C30 VLC-HexCer (Appendix A) was noted with unaltered VLC Cer (Appendix A), and there was a decreasing trend of VLC-LacCer (Appendix A). We could not detect a measurable expression of *Elovl4* in BV2 cells (data not shown).

### 3.4. High-Glucose Induced Changes in Acidic and Neutral Sphingomyelinase (SMase) Enzyme Activity in Cultured HREC and BV2 Cells

We next sought to compare SMase activity between HG and control (NG, LG, and Man)-cultured HREC and BV2 cells at 3, 24, and 48 h. In HG-treated HREC, elevated activity of aSMase at 3 and 48 h was noted compared to controls (Figure 8A). However, no significant change was observed in the activity of nSMase (Figure 8C). In BV2 cells, however, increased nSMase activity was observed at 3 and 24 h (Figure 8D), whereas aSMase activity remained unaltered with high glucose (Figure 8B).

### 3.5. Expression of Sphingolipid Metabolism, Proliferative, and Proinflammatory Markers in HG HREC and BV2 Cells

Along with alterations of the sphingolipid profiles in HG HREC, we expected changes in the expression of their metabolic genes, cellular markers of inflammation, and vascular permeability factors. Using qRT-PCR, we studied the transcript levels of relevant marker genes at 3, 24, and 48 h in HREC cultured in NG and HG. Significant increases in the expression of *Serine palmitoyl transferase 1* (*SPTLC1*), *Ceramide synthase 4* (*CERS4*), *Sphingomyelin phosphodiesterase 1* (*SMPD1*), and *Sphingomyelin phosphodiesterase 2* (*SMPD2*) genes involved in the de novo Cer synthesis and SMase pathways, respectively, at 3 and 24 h in HG-cultured HREC were observed (Figure 9A,B). Further, increased expression of *Lactosyl ceramide synthases 5 and 6* (*B4GALT5* and *B4GALT6*), the LacCer biosynthetic genes, at 24 and 48 h in HG-treated HREC (Figure 9B,C) were observed. Increased expression of cell adhesion molecule *Platelet and endothelial cell adhesion molecule 1* (*PECAM1*) (Figure 9A–C) and *Vascular Endothelial Growth Factor A* (*VEGFA*) (Figure 9C) were noted in HG-treated cells. In addition, elevated levels of inflammatory markers (*Interleukin-6, IL-6; Tumor necrosis factor-α, TNF-α*; *Interleukin-18, IL-18*) were observed in HG-treated HREC (Figure 9A–C).

In contrast to HRECs, HG treatment in BV2 cells caused a significant increase in gene expression of *Sptlc2, Sphingosine kinase 1* (*Sphk1*), and *Glucosyl ceramide synthase* (*Gcs*) (Figure 10A–C). In BV2, while there were no changes in vascular permeability genes, a significant increase in proinflammatory markers (*Il6*, *Tnf-α, Il18, and inducible Nitric oxide synthase; iNos)* was noted in HG-treated cells (Figure 10A–C).

## 4. Discussion

Given the involvement of SPLs in a wide array of cellular functions, it follows logically that differences in the functional niches of different types of cells may be reflected by differences in their SPL composition and metabolism. SPLs are important not only as membrane components but also as mediators of numerous biological processes associated with specialized cell populations. In this study, we performed comprehensive SPL profiling of cultured immortalized retinal endothelial cells (HREC) and microglial cells (BV2), two distinct cell types which fill very different roles but whose concerted activity is tied to the pathophysiology of microvascular diseases such as diabetic retinopathy. We further analyzed how cellular stress from high glucose concentrations influences their SPL composition, SPL metabolic activity, and expression of genes tied to SPL regulation, vascular permeability and proliferation, and inflammation.

We observed differences in SPL levels between HREC and BV2 cells in their respective cultured media conditions. In HREC, the levels of Cer and LacCer were higher than BV2 cells, while BV2 cells showed increased levels of HexCer compared to HREC (Figure 1A,B). HREC also showed approximately 2.5-fold elevated Cer relative composition compared to BV2 cells, while the BV2 HexCer relative composition was about 4-fold higher than HREC (Figure 1C,D). These differences in SPL composition could suggest that the roles of SPLs in cellular events may differ between cell types depending on their lineage or differences in their functional roles for that particular cell type. However, given that HREC are human-derived and BV2 cells are mouse-derived, we cannot rule out the possibility that these differences may reflect the differences in the species of origin. Therefore, the use of human microglial cell lines such as HMC3 (CRL-3304, ATCC) would have been a better choice to compare with HREC. However, considering the fact that lipidomes segregate more widely according to organs rather than species origin, we hypothesize that the lipid profiles of endothelial cells would match more closely to endothelial cells of other origins, and similarly, the lipidome of microglial cells with microglial cells of other origins [34]. A lipidomic study of 6 tissues from 32 mammalian species revealed that while genomic differences contribute to >80% of phylogenetic differences, lipidome’s contribution was only 1.9%, indicating that lipid concentrations evolve differently compared with genome sequences. Thus, the differences between HREC and BV2 are more likely due to their functional changes rather than their species of origin [34].

HREC cells showed elevated levels of LacCer and reduced SM in high-glucose conditions (Figure 3 and Figure 4), while levels of Cer remained unaltered. From the biochemical (lipidomic, SMase) and gene expression assays, it appears that these changes in the SPL profile might result from changes in SMase and LacCer synthase activity but not from the de novo or salvage pathway of Cer synthesis. Cer is a central intermediate of cellular SPL biosynthesis and generates potent signals for apoptosis and inflammation [2,3,9]. Therefore, the levels of Cer species in a cell are tightly regulated and strongly influence cell fate [5]. Although Cer levels were unchanged in glucose-stressed HREC cells, the presence of increased LacCer levels suggests conversion to forms of Cer that do not exude as strong a proapoptotic signal. We also observed increased Cer:SM and Cer:Sph ratios in high-glucose-cultured HREC (Figure 6A,C), while the S1P:Sph ratio remained unaltered (Figure 6E). These elevations of LacCer and changes in Cer:SM and Cer:Sph ratios suggest that high glucose induces changes in the SPL metabolic gradient in endothelial cells. Our earlier studies on human cadaver vitreous samples showed significantly elevated levels of Cer, LacCer, and SM from diabetic patients [35], indicating that SPL alterations could be a characteristic of DM pathology in the eye. In mouse models of STZ-induced Type 1 DM, there are decreased rates of cellular respiration and defects in the calcium retention capacity of mitochondria in cardiac tissue. These changes were associated with elevation of LacCer and could suggest an involvement of glycosphingolipids in diabetic cardiovascular pathology. In the same study, authors also reported increased expression of de novo Cer biosynthetic enzymes, Sptlc1 and CerS2, in cardiac tissue. Elevations of LacCer in knockout mouse models of neutral ceramidase deficiency further suggest a role of Cer and LacCer in the pathogenesis of glucoregulatory disorders [36]. We also demonstrated increased transcript levels of SPTLC1 and CERS2 in high-glucose-cultured cells (Figure 9), which further supports associations between glucose-induced cell stress and changes in Cer and LacCer metabolism. LacCer activation mediates signaling pathways that modulate cell proliferation, adhesion, migration, and angiogenesis [37]. Endothelial dysfunction in proliferative DR is associated with the generation of proangiogenic factors, namely, VEGF, which stimulates pathological neovascularization [38]. VEGF-mediated angiogenic activity with concomitant elevation of PECAM1 is a signature of neovascularization and may be associated with changes in LacCer in endothelial cells, suggesting that LacCer synthase could be a downstream effector of VEGF-induced angiogenesis [39,40]. In agreement with this, we observed increased transcript levels of VEGFA (Figure 9), LacCer synthase genes (*B4GALT5* and *B4GALT6*), and *PECAM1* (Figure 9) in high-glucose-cultured HREC. This correlation suggests the possibility that LacCer could be explored as a potential prognostic or diagnostic marker in hyperglycemia and DR, which may warrant further study.

Along with the increase in LacCer, we noticed a 10% reduction in SM in HG-treated cells (Figure 3). SM is the most abundant SPL in any cell and the third most abundant SPL in the cell membrane and constitutes the majority of lipid rafts [5]. Changes in SM composition can significantly affect cell fate and signaling as this can change the biophysical properties of the lipid rafts and thus can influence the activity and functions of many proteins (enzymes and channels) associated with lipid rafts. Cellular stoichiometry of SM-Cer is maintained so that a 1% change in SM composition may cause a 30% change in Cer; Cer is a bioactive lipid and is well known for signaling inflammatory and apoptotic pathways [2,5]. Alternatively, Cer can also be converted to C1P or Sph, followed by S1P; all of them can have signaling roles. The reduction in SM in HG-treated cells could also be associated with increased aSMase activity, which acts either in lysosome or on the outer leaflet of the cell membrane after vesicular transport [41]. Finally, SMase activation and generation of Cer are known to be one of the major pathways by which TNFα induces cellular inflammatory signaling [2,5,42]. Thus, SMase activation in HG-treated cells and SM reduction may have important biological significance. Future studies are warranted to understand better the changes in LacCer and SM influencing cell fate and biophysical properties of the lipid rafts under high-glucose stress.

Angiogenic factors play a major role in disrupting the blood–retinal barrier (BRB) in DR. High glucose stimulates the formation of microcapillaries by destabilizing tight junctions with the help of inflammatory factors [43]. The elevated activity of aSMase in endothelial cells in diabetic individuals increases Cer synthesis, which helps disrupt retinal vasculature and induce retinal inflammation [18,19]. We observed higher enzymatic activity of aSMase (Figure 8A) and elevated transcript levels of proinflammatory cytokines (IL-6, IL-18, and TNF-α) (Figure 9) in high-glucose-cultured HREC, which mimics the prevailing phenotypic characteristics observed in diabetic retinas. Likewise, in STZ-induced diabetic rat retinas, the breakdown of the BRB was accelerated by an aSMase-induced accumulation of Cer, which is associated with increases in proinflammatory cytokines IL-1β and IL-6 [44]. There is a delicate balance between short-chain Cer species, proinflammatory, and VLC Cer species, which help maintain tight junctions in the retina [45]. In the diabetic retina, this delicate balance is tilted towards higher levels of short-chain Cer species, contributing to compromised vascular integrity [33]. The decreased levels of VLC Cer might be related to the downregulated activity of ELOVL4 [33]. High-glucose treatment significantly downregulated the expression of *ELOVL4* (Figure 7E) in HREC, resulting in the accumulation of C26:1 and C26:0 LacCer (Figure 7D) and SM (Figure 7B), and a reduction in VLC (C28-C30) Cer, LacCer, and SM species (Figure 7A,B,D). A similar downregulation of *Elovl4* expression was observed in STZ-induced diabetic rat retinas [46]. Overexpression of ELOVL4 in cultured bovine endothelial cells was shown to restrict basal permeability and permeability induced by IL-1β and VEGF [33]. VLC SPL-mediated alteration of VEGF may affect tight junction protein ZO1 and claudin [33]. In the present study, we observed decreased VLC Cer, LacCer, and SM levels associated with increased transcript levels of VEGFA and PECAM1. Our observations of reduced levels of VLC LacCer in HG-treated HREC are supported by findings from the Diabetes Control and Complications Trial (*DCCT*)/Epidemiology of Diabetes Interventions and Complications (EDIC) Type 1 diabetes sub-cohort, which sought to identify glycosphingolipids as potential biomarkers in diabetic complications [47]. The study reported significantly decreased plasma VLC-LacCer levels associated with macroalbuminuria in Type 1 DM [47]. Thus, ELOVL4 and VLC SPL species might have specific roles in vascular abnormalities in diabetes, though further investigation is needed.

In BV2 cells, we observed elevations of HexCer (Figure 5) and S1P (Appendix A) and a significant reduction in SM (Figure 5). Unlike HREC, nSMase activity was increased in HG-treated BV2 cells but with no change in aSMase levels (Figure 8), which might have contributed to the decrease in SM. Additionally, the Cer generated might have been converted to HexCer to maintain cellular homeostasis by restricting Cer elevation [5]. The increase in expression of the *Gcs* gene may support that GCS activity might have been increased in HG-treated BV2 cells (Figure 10). Similarly, higher expression of *Sphk1* (Figure 10) may support the higher activity of SPHK1 protein and increased levels of S1P in HG-treated BV2 cells. STZ-induced Type 1 diabetic rat retinas exhibited significantly increased levels of GlcCer and decreased levels of Cer, which could possibly induce endoplasmic reticulum stress, as evidenced by increased expression of Glucose-related protein 78 (GRP78) and chaperone protein CHOP [15]. Higher levels of short-chain HexCer (C14:0 and C18:0) in HG-induced BV2 cells (Appendix A) and C16:0 LacCer in HREC (Appendix A) follow a similar trend of pathological elevation of C16:1 and C18:2 GlcCer in the retinas of diabetic rat models [15]. Lipidomic studies of plasma in T2DM patients have also shown associations between short-chain C16:1 and C18:2 HexCer and characteristics of obesity vis-à-vis DM [48]. Earlier, we also reported higher levels of HexCer in diabetic vitreous samples [35], potentially supporting a pathological role of glycosphingolipids in DM. Glycosphingolipids, both GluCer and LacCer, are associated with plaque-related inflammation, and their significantly elevated presence in atherosclerotic plaques supports their pathological role in this regard [49,50]. GluCer and LacCer may also act as neuroinflammatory agents in Parkinson’s disease, supporting their role as inflammatory effectors within the CNS [51]. These findings highlight the potential role of SPLs in acting as stimuli or intermediate agents in inflammatory processes. Inflammation is one of the major causes of pathological changes in diabetic patients [52], and microglial cells play a critical role in inflammation within the CNS and the retina [20]. Other in vitro studies of high-glucose-cultured BV2 cells showed increased levels of inflammatory factors [53], while blocking the toll-like receptor-4/nuclear factor kappa-B pathway inhibits high-glucose-induced inflammation in BV2 cells [54], suggesting that elevation of glucose acts as a proinflammatory stimulus for microglia. Experimental models mimicking diabetic pathology using high-glucose/high-cholesterol conditions in zebrafish also showed the involvement of microglia and secretion of inflammatory cytokines [55]. Our observed elevations of inflammatory cytokine transcription in high-glucose-cultured HREC and BV2 cells also support a potential pathological effect of glycosphingolipids. Plasma samples of experimental Type 1 diabetic rats and Ins2^Akita^ diabetic mice have shown significant elevations of bioactive S1P [56]. S1P acts as an agent for generating normal retinal vasculature in the retina. It also triggers the secretion of inflammatory cytokines and proliferating factors that can destabilize retinal vasculature [9]. The increased ratios of Cer:SM, Cer:Sph, and S1P:Sph (Figure 6B,D,F) in BV2 cells could suggest the conversion of S1P, while the increased transcript levels of Sphk1 (Figure 10A,C) indicate the elevation of S1P synthesis in high-glucose conditions.

Omics studies of gene and protein expression provide helpful information for understanding tissue function. With the recent advances in shotgun lipidomics, the quantification of hundreds of lipids from a sample presents a very useful tool for correlative studies of gene and protein expression in understanding tissue function in normal and disease conditions [57]. The present study suggests that sphingolipid profiles may differ between cell populations based on their specialized functional roles, and that changes in sphingolipid metabolism induced by stressors may differ based on the identity or origin of the affected cell. In the present study, we noticed striking differences in SPL profiles between HREC and BV2 cells. In high-glucose-treated HREC, we observed alterations in levels of LacCer and VLC SPL, as well as changes in cell proliferative and proinflammatory factor gene transcription. Under similar conditions, BV2 cells showed elevations of HexCer and S1P levels and changes in proinflammatory factor transcription but no changes in proliferative factor gene transcription (Figure 9 and Figure 10). This in vitro pilot study suggests that alterations in SPL composition and metabolism are tied to the distinct responses of specialized cell types to high-glucose-induced stress, which could further imply that SPLs are involved in modulating critical processes specific to these cells. High-glucose-mediated lipid toxicity and altered metabolism are strongly implicated in the pathogenesis of DM and diabetes-related complications [11]. Hyperglycemic metabolic stress can trigger proinflammatory phenotypic changes in various cell types, including endothelial cells and immune cells, which are believed to contribute to macro- and microvascular pathology associated with DM. The interplay between these cell types is believed to contribute to the pathogenesis of DR, where chronic glucose-induced cellular stress leads to the development of retinal lesions, vascular abnormalities, and neuronal atrophy [58]. The endothelial cells lining the inner lumens of retinal blood vessels regulate vascular permeability, blood flow, and nutrient exchange, play a critical role in neovascularization (NV), and help modulate the activity and movement of immune cells. Long-term exposure to elevated glucose levels leads to chronic, low-level activation of inflammatory signaling in endothelial cells, causing changes to vascular permeability, endothelial proliferation and pathologic neovascularization, thickening of basement membranes, and loss of pericytes [59]. Vasculopathy in DR is also associated with microglial morphological changes such as process shortening and decreased ramification, which are consistent with inflammatory phenotype switching. The presence of microglial perivasculitis in DR highlights the interplay between endothelial and microglial cells in glucose-induced inflammation and vascular pathology [25]. Given the involvement of SPL signaling in inflammation and neovascularization, it is possible that conditions which induce these processes could result in changes in SPL metabolism in cells that take an active role in them.

Taken together, the present study highlights the potential involvement of SPLs in high-glucose-mediated metabolic changes that could act as prognostic or/and therapeutic markers in different pathological complications, including diabetes. However, one limitation of our study is that our research does not present a causal connection between the changes in SPL profiles under high-glucose stress and SPL profile changes to observed cellular phenotypes. Further studies are needed to genetically (such as using siRNA) or pharmacologically inhibit one pathway or enzyme at a time and dissect the individual pathways or the enzyme’s causative role with the changes observed in the SPL profile or the phenotype of the cells.

## Figures and Tables

**Figure 1 cells-11-03082-f001:**
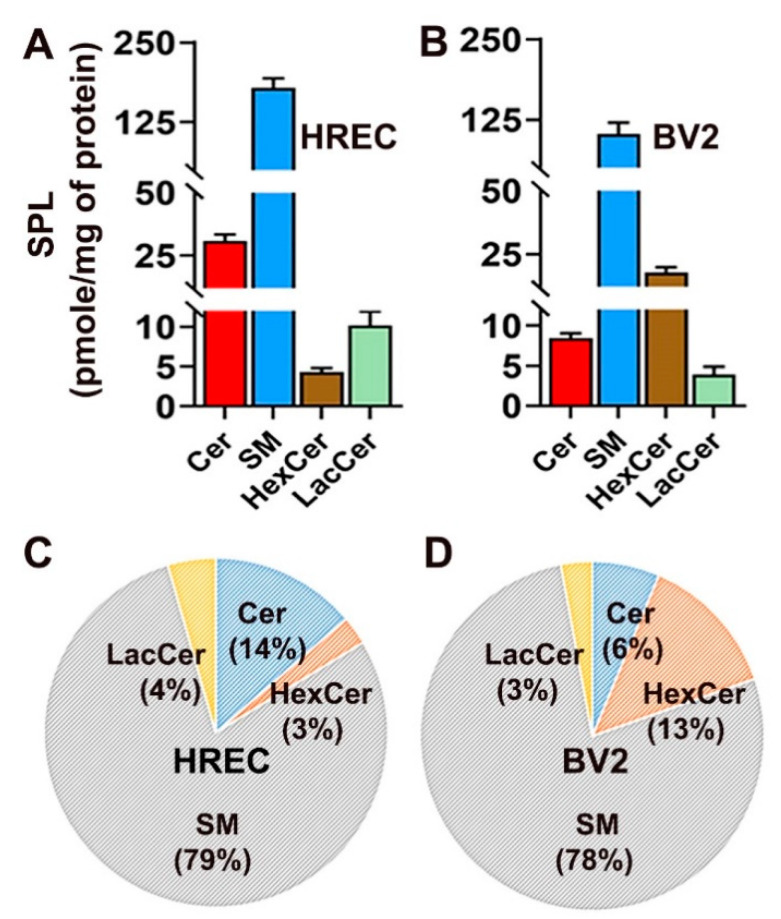
Analysis of major SPL in cultured human retinal endothelial cells (HREC) and murine microglial cells (BV2). Concentration (pmol/mg of protein) of Cer, SM, HexCer, and LacCer in (**A**) HREC and (**B**) BV2 cells cultured for 48 h in 5 mM glucose-containing media. Compositional analysis (mol%) of SPL species (**C**) HREC and (**D**) BV2 cells. Mean ± SEM; *n* = 3.

**Figure 2 cells-11-03082-f002:**
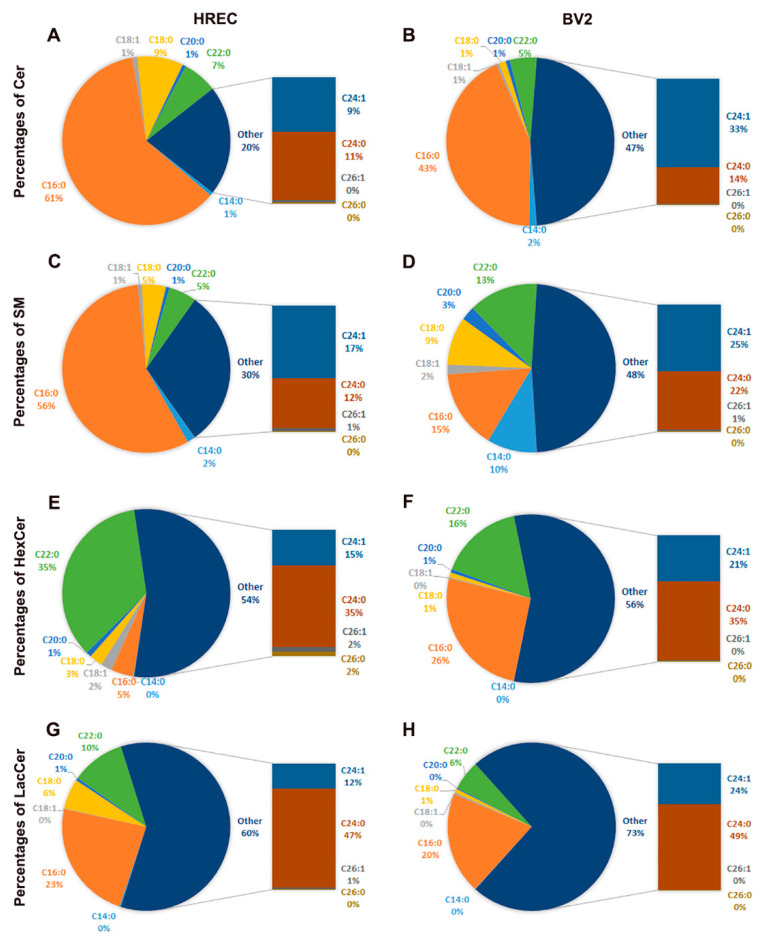
Compositional analysis of major SPL carbon chain length variants in HREC and BV2 cells cultured for 48 h in 5 mM glucose-containing media. Mol% composition of (**A**) Cer, (**C**) SM, (**E**) HexCer, and (**G**) LacCer species in HREC. Mol% composition of (**B**) Cer, (**D**) SM, (**F**) HexCer, and (**H**) LacCer species in BV2 cells. *n* = 3.

**Figure 3 cells-11-03082-f003:**
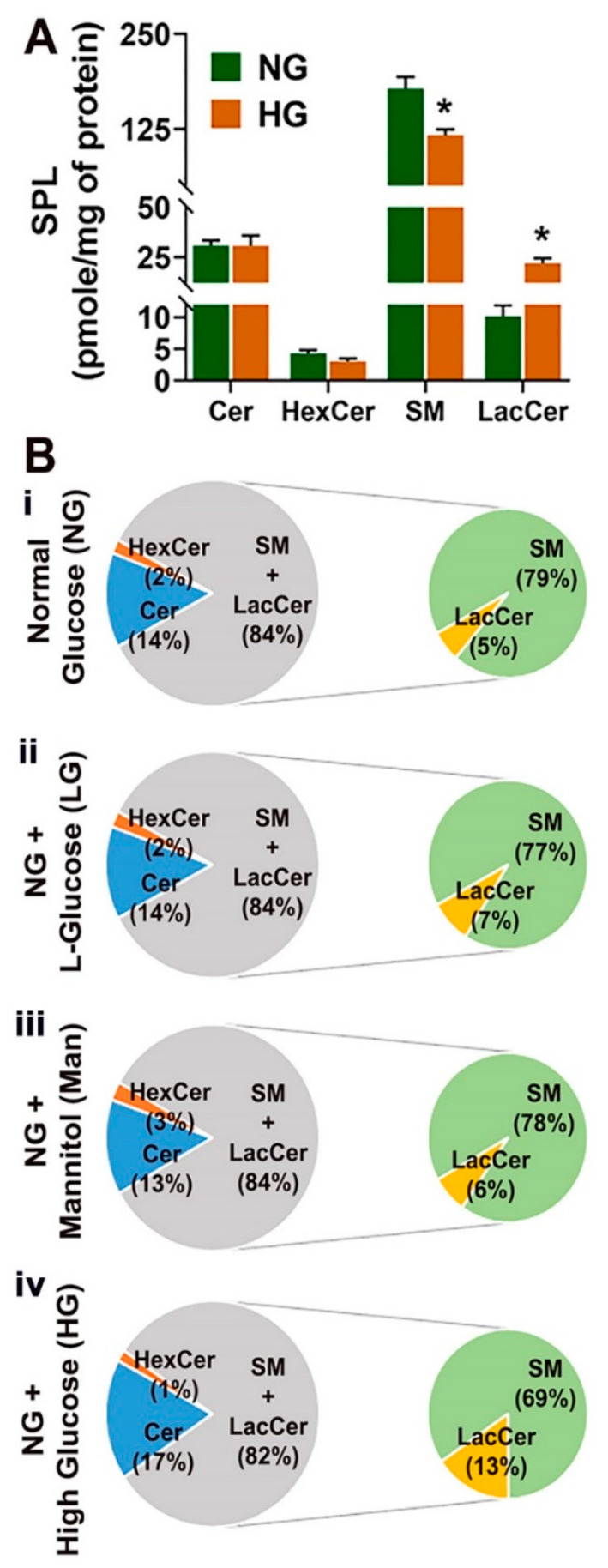
Analysis of major SPL classes in HREC cultured in 5 mM glucose (NG), 25 mM glucose (HG), NG + 20 mM L-glucose (LG), and NG + 20 mM mannitol (Man)-containing media for 48 h. (**A**) Total concentration (pmol/mg protein) of Cer, HexCer, SM, and LacCer in NG and HG culture conditions. (**B**) Compositional analysis (mol%) of Cer, SM, HexCer, and LacCer in (**i**) NG, (**ii**) LG, (**iii**) Man, and (**iv**) HG-cultured HREC. * *p* < 0.05; mean ± SEM; *n* = 3.

**Figure 4 cells-11-03082-f004:**
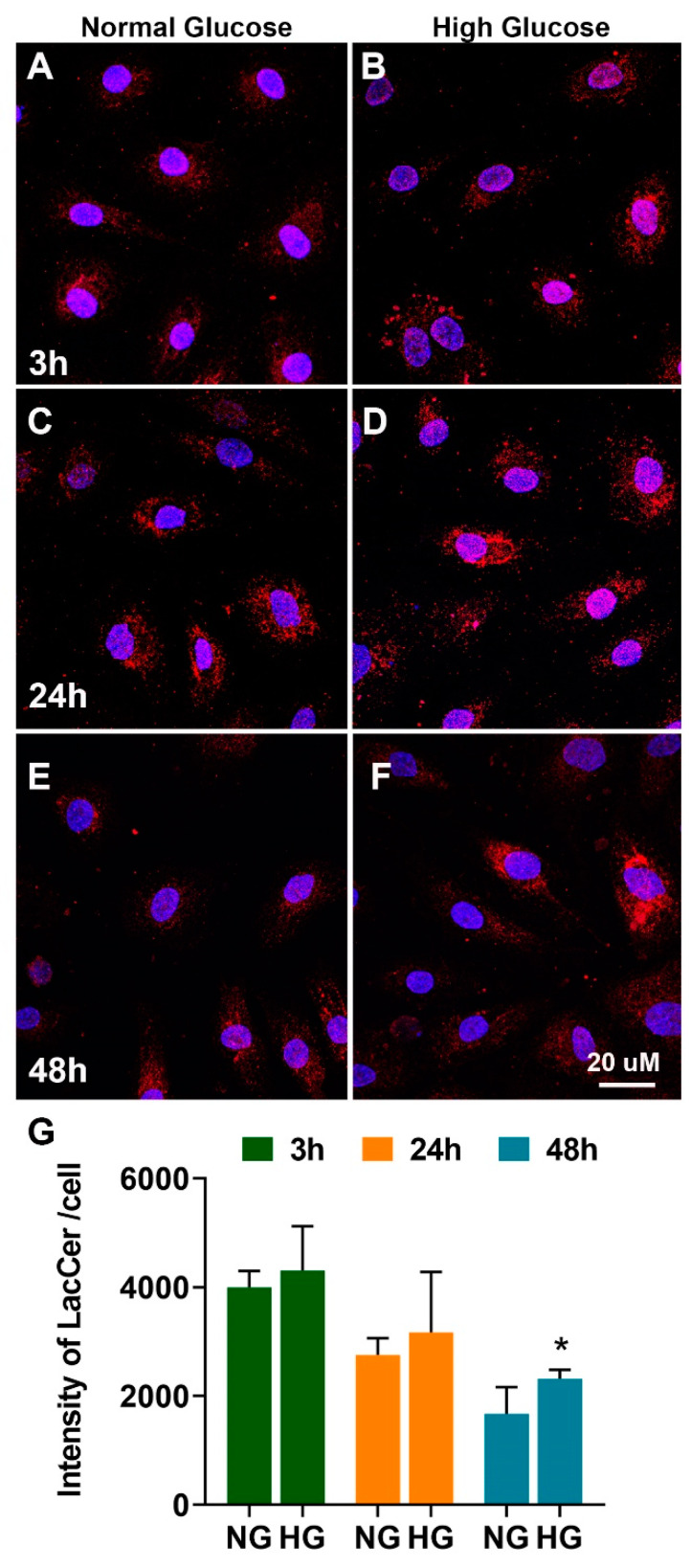
Immunohistochemical staining of LacCer in HREC cultured in 5 mM glucose (normal glucose) or 25 mM glucose (high-glucose) media for 3, 24, and 48 h. Confocal micrographs show LacCer (CD17) in red and nuclei (DAPI) in blue (**A**–**F**). LacCer labeling was quantified by ImageJ analysis and expressed as intensity/cell from 5–6 fields per condition and from at least 3 independent experiments (**G**). * *p* < 0.05.

**Figure 5 cells-11-03082-f005:**
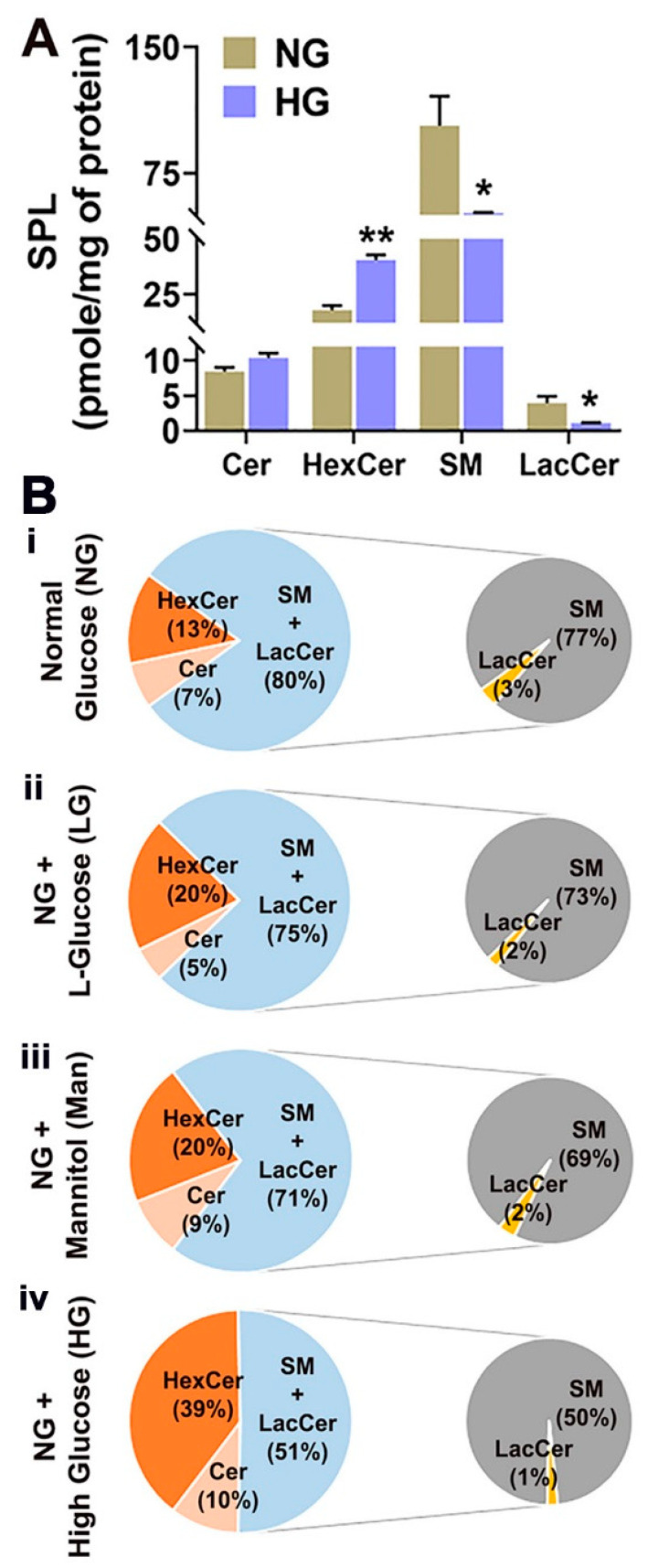
Analysis of major SPL classes in BV2 cells cultured in 5 mM glucose (NG), 25 mM glucose (HG), 5 mM glucose + 20 mM L-glucose (LG), and 5 mM glucose + 20 mM mannitol (Man)-containing media for 48 h. (**A**) Total concentration (pmol/mg protein) of Cer, HexCer, SM, and LacCer in NG and HG culture conditions. (**B**) Compositional analysis (mol%) of Cer, SM, HexCer, and LacCer in (**i**) NG, (**ii**) LG, (**iii**) Man, and (**iv**) HG-cultured HREC. * *p* < 0.05, ** *p* < 0.01; mean ± SEM; *n* = 3.

**Figure 6 cells-11-03082-f006:**
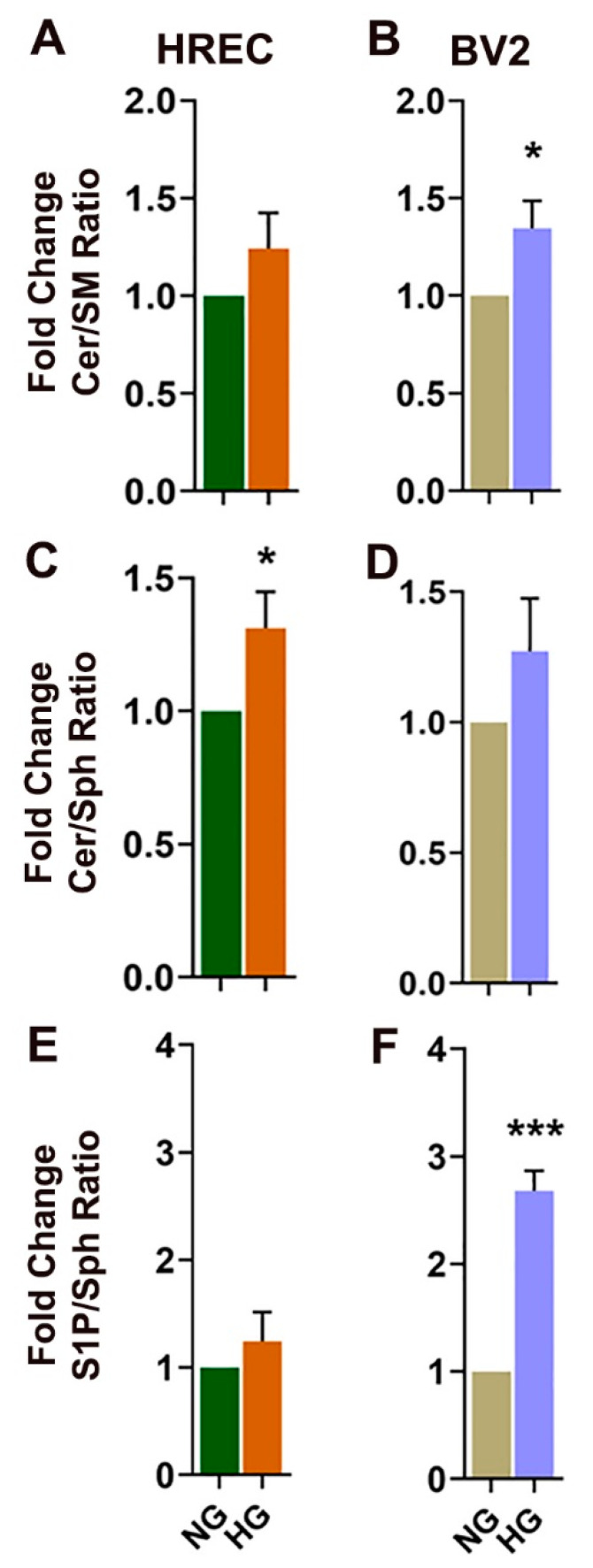
Ratios of major SPL classes in HREC and BV2 cells cultured in 5 mM glucose (NG) or 25 mM glucose (HG) conditions for 48 h. (**A**) Cer:SM, (**C**) Cer:Sph, and (**E**) S1P:Sph ratios in HREC. (**B**) Cer:SM, (**D**) Cer:Sph, and (**F**) S1P:Sph ratios in BV2 cells. * *p* < 0.05, *** *p* < 0.001; *n* = 6.

**Figure 7 cells-11-03082-f007:**
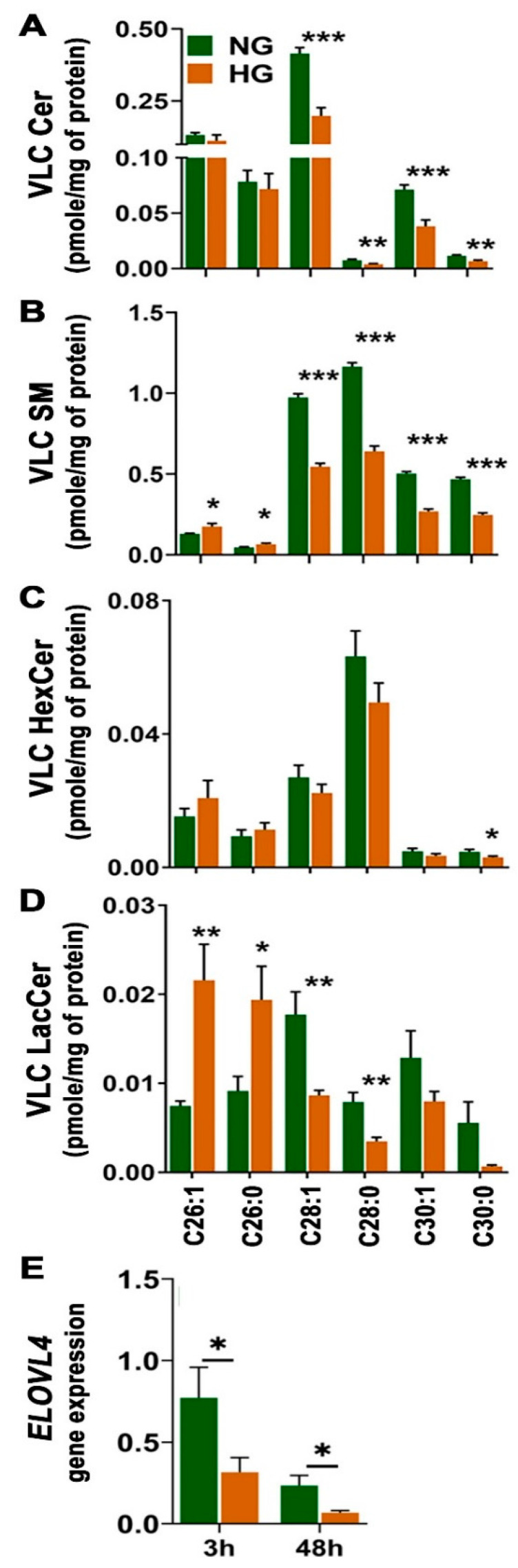
Differences in major VLC SPL levels and *ELOVL4* gene expression in HREC cultured in 5 mM glucose (NG) or 25 mM glucose (HG) conditions. Levels of (**A**) VLC Cer, (**B**) VLC SM, (**C**) VLC HexCer, (**D**) VLC LacCer in HREC cultured in NG or HG for 48 h (pmol/mg protein; * *p* < 0.05, ** *p* < 0.01, *** *p* < 0.001; *n* = 6). (**E**) RT-qPCR gene expression analysis of *ELOVL4* in HREC cultured in NG or HG media for 3 and 48 h (mean ± SEM; * *p* < 0.05; *n* = 3).

**Figure 8 cells-11-03082-f008:**
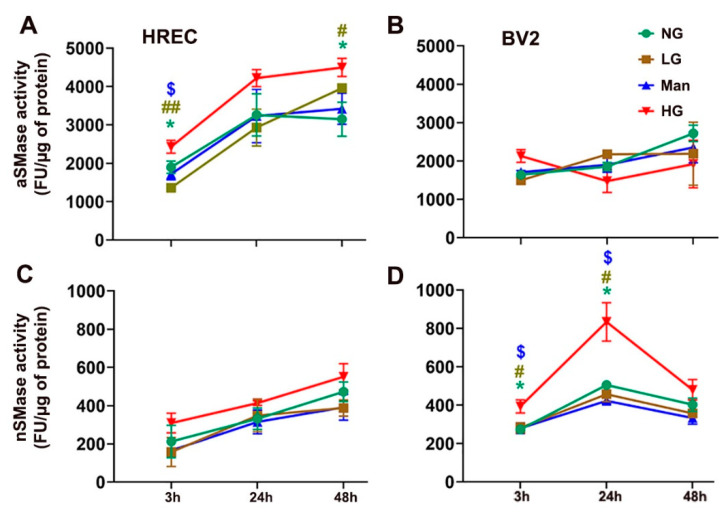
Analysis of SMase enzymatic activity (fluorescence unit/µg of protein) in HREC and BV2 cells cultured in normal glucose (NG), NG + L-glucose (LG), NG + mannitol (Man), and high-glucose (HG) conditions for 3, 24, or 48 h. (**A**) aSMase and (**C**) nSMase activity in HREC. (**B**) aSMase and (**D**) nSMase activity in BV2 cells. * *p* < 0.05 LG vs. HG; ^#^
*p* < 0.05, ^##^
*p* < 0.01 Man vs. HG; ^$^
*p* < 0.05 NG vs. HG; mean ± SEM; *n* = 3.

**Figure 9 cells-11-03082-f009:**
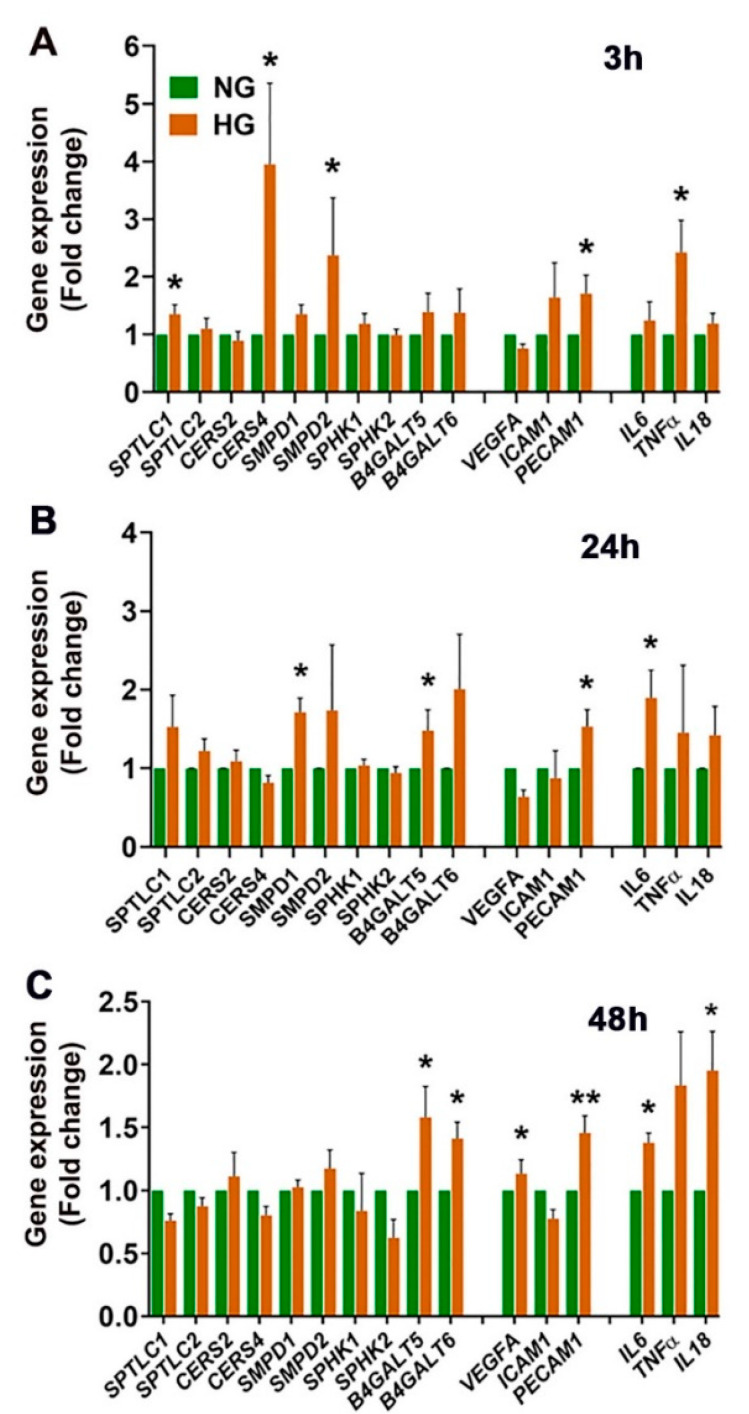
RT-qPCR gene expression analysis of sphingolipid metabolism, vascular proliferation, and inflammatory marker genes in HREC cultured in 5 mM glucose (NG) or 25 mM glucose (HG) conditions for 48 h. Gene expression changes represented as fold change over control (= 1.0) following normalization with two housekeeping genes *RPL19* (*ribosomal protein L19*) and *GAPDH* (*Glyceraldehyde-3-phosphate dehydrogenase)* after cells were cultured for (**A**) 3, (**B**) 24, and (**C**) 48 h in NG and HG. *SPTLC1*, *Serine palmitoyl transferase 1*; *SPTLC2*, *Serine palmitoyl transferase 2;*
*CERS2*, *Ceramide synthase 2;*
*CERS4*, *Ceramide synthase 4*; *SMPD1, Sphingomyelin phosphodiesterase 1*; *SMPD2, Sphingomyelin phosphodiesterase 2; SPHK1, Sphingosine kinase 1*; *SPHK2, Sphingosine kinase 2*; *B4GALT5, Lactosyl ceramide synthase 5*; *B4GALT6*, *Lactosyl ceramide synthase 6*; *VEGFA*, *Vascular endothelial growth factor A*; *ICAM1*, *Intracellular adhesion molecule 1*; *PECAM1*, *Platelet endothelial cell adhesion molecule 1; IL-6*, *Interleukin 6*; TNF-α, *Tumor Necrosis α*; *IL-8*, *Interleukin 18*. * *p* < 0.05, ** *p* < 0.01; *n* = 6.

**Figure 10 cells-11-03082-f010:**
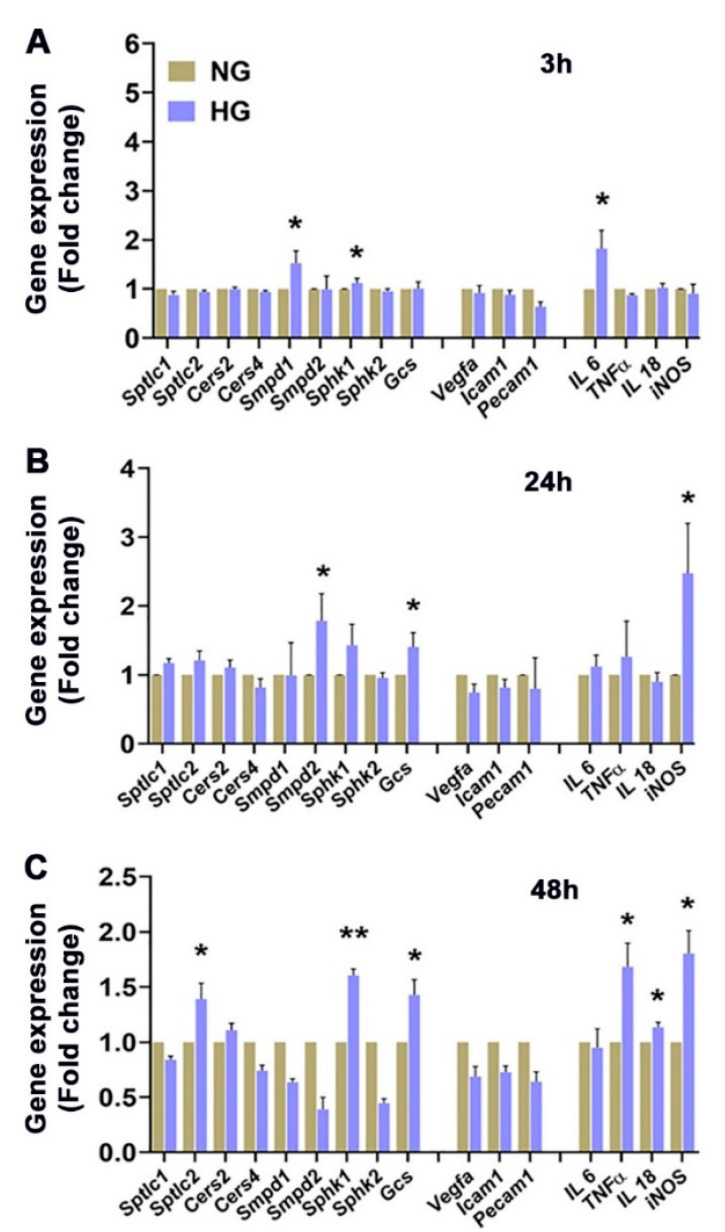
RT-qPCR gene expression analysis of sphingolipid metabolism, vascular proliferation, and inflammatory marker genes in BV2 cells cultured in 5 mM glucose (NG) or 25 mM glucose (HG) conditions for 48 h. Gene expression changes represented as fold change over control (=1.0) following normalization with two housekeeping genes *Rpl19* and *Gapdh* after cells were cultured for (**A**) 3, (**B**) 24, and (**C**) 48 h in NG and HG. *Sptlc1*, *Sptlc2*, *Cers2*, *Cers4*, *Smpd1*, *Smpd2*, *Sphk1, Sphk2*, *Glucosyl ceramide synthase* (*Gcs*); *Vegfa*, *Icam1*, *Pecam1, Il6*, *Tnfα*, *Il18*, *inducible Nitric oxide synthase* (*iNOS*). * *p* < 0.05, ** *p* < 0.01; *n* = 6.

## Data Availability

All materials and data will be available from the corresponding author following University of Tennessee’s policy of sharing research materials and data.

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
