# Peer review of "A Comprehensive Profiling of Cellular Sphingolipids in Mammalian Endothelial and Microglial Cells Cultured in Normal and High-Glucose Conditions"

_cells, 2022, doi:10.3390/cells11193082_

Round 1

Reviewer 1 Report

The authors demonstrate that normal (5 mM; NG) or high (20 mM; HG) glucose culture conditions differentially affect sphingolipid (SL) concentrations and composition in human retinal endothelial cells (hREC) and in a murine microglia cell line (BV2). A potential relationship between diabetic retinopathy and glucose-induced changes in SL species composition, SMPD1/2 activity, and expression of a selected set of gene products (involved in SL synthesis, angiogenesis, and the inflammatory response) is discussed.

Although species difference (hREC vs. BV2) is briefly stated (lines 428 ff) a clear rationale for the choice of the murine cell line must be provided. A human microglia cell line (HMC3 - CRL-3304) is commercially available (ATCC).

In the context of species differences it might be worthwhile to mention that lipidomes segregate predominately according to organ- rather than species origin (doi:10.1093/molbev/msy097 Khraameva et al.).

MS analyses: All species separated on a C18 column or combination of C18 and NH2?

Primer sequences for qPCR target genes should be provided in M&M.

Fig. 3: Increase in LacCer due to increased de novo synthesis or decreased salvage pathway activity?

Fig. 4: Immunofluorescence - secondary antibody? Please provide statistical evaluation, in which compartment is LacCer detected?

The authors should briefly discuss possibilities to pharmacologically/genetically (siRNAs) interfere with SL synthesis pathways to obtain direct and causative evidence that alterations in SL homeostasis are responsible for the observed phenotypes.

I suggest alleviating the statement in line 527 and to discuss doi: 10.1016/j.celrep.2020.108132 (Fitzner et al.).

Author Response

The authors demonstrate that normal (5 mM; NG) or high (20 mM; HG) glucose culture conditions differentially affect sphingolipid (SL) concentrations and composition in human retinal endothelial cells (hREC) and in a murine microglia cell line (BV2). A potential relationship between diabetic retinopathy and glucose-induced changes in SL species composition, SMPD1/2 activity, and expression of a selected set of gene products (involved in SL synthesis, angiogenesis, and the inflammatory response) is discussed.

 Although species difference (hREC vs. BV2) is briefly stated (lines 428 ff) a clear rationale for the choice of the murine cell line must be provided. A human microglia cell line (HMC3 - CRL-3304) is commercially available (ATCC).

In the context of species differences it might be worthwhile to mention that lipidomes segregate predominately according to organ- rather than species origin (doi:10.1093/molbev/msy097 Khraameva et al.).

We thank the reviewer for raising this critical point. The choices of cells (HREC and BV2) were serendipitous and purely based on the availability of those cells in the lab and our familiarity with culturing them. We indeed observed marked differences between these cell lines on their culture conditions, growth patterns, etc., and we always attributed that to their characteristics of 'endothelial cells' vs. 'microglial cells.' We wish we were right in our assumption. However, we are also baffled by the question, especially after years of efforts, completing sphingolipidomic studies, and observing significant differences in their patterns, whether this reflects species differences (human vs. rodent). Therefore, we agree with you that we must do a comparative study using a human microglia cell line (e.g., HMC3 - CRL-3304); however, this work is beyond the scope of current work to establish the SL composition across cell types under various glucose stress cnditions. We have incorporated your helpful suggestion that lipidomes segregate predominately according to organ- rather than species origin (See discussion, lines 457-467). We have also discussed the rationale better to explain why our existing data is informative in demonstrating differences in the nature of the cells (endothelial vs. microglial) as well as discussed the limitations of our study (See discussion, lines 636-639).

MS analyses: All species separated on a C18 column or combination of C18 and NH2?

A combination of C18 and LC-NH2 columns were used to analyze all species of sphingolipids following the methodology of Dr. Alfred Merill [Shaner RL, Allegood JC, Park H, Wang E, Kelly S, Haynes CA, Sullards MC, Merrill AH Jr. Quantitative analysis of sphingolipids for lipidomics using triple quadrupole and quadrupole linear ion trap mass spectrometers. J Lipid Res. 2009 Aug;50(8):1692-707. doi: 10.1194/jlr.D800051-JLR200. Epub 2008 Nov 25. PMID: 19036716; PMCID: PMC2724058.]. We have updated this information in the methods section (Lines 138-139).

Primer sequences for qPCR target genes should be provided in M&M.

We have provided Primer sequences of all the target genes in the Supplementary materials as Supplementary Tables 1 and 2.

Fig. 3: Increase in LacCer due to increased de novo synthesis or decreased salvage pathway activity?

We thank the reviewer for raising this critical point. In HREC, we did not observe a change in the Cer levels or HexCer levels but a reduction in SM and an increase in LacCer. Along with those, the SMase assay and gene expression data indicate that LacCer increase is not controlled at the Cer level i.e., neither by the de novo nor the salvage pathway. It could therefore be by the increase and activation of SMase and LacCer synthase. Consequently, we have modified the discussion to reflect this point. Please see lines 469-473.

Fig. 4: Immunofluorescence - secondary antibody? Please provide statistical evaluation, in which compartment is LacCer detected?

The secondary antibody for the Fig. 4 immunofluorescence was Alexa Fluor 568 (Invitrogen catalog # A11037). This information is now included in the revised manuscript. We also modified Figure 4, added quantitative data on the labeling, and edited the results section to reflect these modifications. Based on enzymatic compartmentalization, we believe the LacCer accumulation is occurring in the Golgi apparatus; however, more studies are warranted to confirm our hypothesis.

The authors should briefly discuss possibilities to pharmacologically/genetically (siRNAs) interfere with SL synthesis pathways to obtain direct and causative evidence that alterations in SL homeostasis are responsible for the observed phenotypes.

We thank the reviewer for this suggestion. We have added the suggested discussion in the revised manuscript (Line 633-639).

I suggest alleviating the statement in line 527 and to discuss doi: 10.1016/j.celrep.2020.108132 (Fitzner et al.).

We did shorten the suggested statement and added a discussion from Fitzner et al. 2020. Line 595-599.

Reviewer 2 Report

The manuscript by Mondal et al. examines changes in sphingolipids in two mammalian cell lines under normal and high glucose conditions.

The lipid analytic is technically sound. The manuscript, however, is largely descriptive and does not give much information on the signalling pathway(s) mediating the high-glucose effect.

Specific Comments

Figure 4:

LacCer immunostaining showed that increased staining is apparently not because of an incerase in all LacCer positive membranes but due to strong increase in some structures that are not further characterized: what are these structures? Costaining with markers for different organelles would be very helpful (this could also ensure that these strongly stained structures are not any staining artifacts.

Figure 6:

Apparently the ratios for the NG controls was always set to 1 in Figure 6. How was the significance test calculated (according to the methods it should be done by ANOVA). I think ANOVA would not be the correct test, if the ratio of all NG controls was set to 1; or was the calculation done with un-normalized data? In that case, showing the un-normalized data would be more approbiate.

Figures 9 and 10:

Same comments as for Figure 6 (see above)

Minor comments:

lines 109, 217, 294, 321: "pmole" should be "pmol"

line 176: "Mean" should be "mean"

Author Response

The manuscript by Mondal et al. examines changes in sphingolipids in two mammalian cell lines under normal and high glucose conditions.

The lipid analytic is technically sound. The manuscript, however, is largely descriptive and does not give much information on the signalling pathway(s) mediating the high-glucose effect.

We thank the reviewer for the encouragement and insightful comment. We agree that our current study is descriptive and focuses on the characterization of sphingolipid synthesis and the differences between cell types and culture conditions under high glucose stress. However, our study, for the first time, provides valuable information on sphingolipid metabolism and signaling in two distinct cell types of the retina, significantly improving our knowledge and understanding of SPL metabolism and signaling that has received very little focus in the past. Our future efforts will focus on signaling pathways based on the significant findings presented in this manuscript. For example, we are pursuing further studies to understand how increased LacCer is related to glucose response in retinal endothelial cells.

Specific Comments

Figure 4:

LacCer immunostaining showed that increased staining is apparently not because of an incerase in all LacCer positive membranes but due to strong increase in some structures that are not further characterized: what are these structures? Costaining with markers for different organelles would be very helpful (this could also ensure that these strongly stained structures are not any staining artifacts.

We agree with the reviewer that costaining for different organelles might be helpful. However, unfortunately, we did not use any organelle markers for costaining studies and, therefore, are unable to address this question. To reduce the bias in interpreting our existing data (see a comment from Reviewer 3), we have captured images at lower magnification to include several cells per field, quantified the labeling, and revised the figure. Our data confirm our previous observation that LacCer increases under high glucose stress (at 48h). We firmly believe and expect the reviewer to agree that our re-analysis has significantly reduced the selection bias or staining artifacts, if any. Please see the result section to reflect the modifications and the revised Figure 4.

Figure 6:

Apparently the ratios for the NG controls was always set to 1 in Figure 6. How was the significance test calculated (according to the methods it should be done by ANOVA). I think ANOVA would not be the correct test, if the ratio of all NG controls was set to 1; or was the calculation done with un-normalized data? In that case, showing the un-normalized data would be more approbiate.

Figures 9 and 10:

Same comments as for Figure 6 (see above)

We apologize for not explaining the data analysis better. The un-normalized data were first analyzed with ANOVA (for more than 2 groups) or t-test (between 2 groups), and the significance (p-value) was determined. As the variability of the expression of different genes (for example) is so high, it is not possible to plot them in one graph. We, therefore, calculated the average of the control group (NG) as 1 and used that to convert each sample of the experimental group (HG). Thus experimental group has a mean and standard error (SE) for presentation but not the control group. However, the statistical significance (*; p-value) was added from the previous calculation (un-normalized data).

Minor comments:

lines 109, 217, 294, 321: "pmole" should be "pmol"

Thank you, it has been corrected in the revised version.

line 176: "Mean" should be "mean"

Thank you, it has been corrected in the revised version.

Reviewer 3 Report

In this paper Mondal and collaborators investigated differences in sphingolipids metabolism between human primary retinal endothelial (HREC) and murine microglial cells (BV2) in normal conditions and under high glucose-induced stress. The manuscript is generally well written and the sphingolipidomics data are interesting and promising. However, at this stage the paper is just descriptive and the results are just a picture of what happens in the two cell lines in the presence of high glucose. The main weakness is the lack of mechanistic insights. Furthermore, a very important weak point is the choice of comparing two cell lines of different origin (human and murine) which, according to the authors, may be at the basis of the differences observed in the SPL profile. Therefore, I have some concerns that should be clarify by the author:

-       The story depicted in the discussion for HREC and neovascularization when cultured in high glucose, is nice, but so far is just a speculation. It should be further validated with silencing experiments of B4GALT5 and B4GALT6, VEGFA and PECAM, to evaluate the pathway suggested. The same applies for the increase of aSMase activity and inflammation and its relationship with the breakdown of the BRB. At this stage is only a speculation, no data is shown that can support this hypothesis. A loss of function experiment (i.e. silencing) in the presence of HG should be performed.

-       Also for BV2 the data are nice and promising, but still preliminary and only descriptive. Moreover,  the authors should speculate more about the different SPL content in the two cell lines investigated. As I said above, if they think that the differences found can be due to the different species origins, why they did not use a human microglial cell line?

-       The images in Fig.4 are not clear. Phalloidin staining of the cytoskeleton to see the morphology of the cells must be performed, otherwise we cannot be sure that the cells are just folded back on themselves. Moreover, showing only one cell per timing is not representative of the situation. A larger field with more cells must be acquired and shown. A quantification of the fluorescence intensity/field must be also performed.

Minor:

-       the second paragraph of introduction should be reorganized. Microglial cells appeared out of the blue (line 58-59) and were then introduced and described later (line 70-71).

-       A 10% reduction of SM in HG-treated HREC cells is biologically relevant? The authors should discuss this issue.

-       Fig. 6: The y-axis label seems to me to be the fold change of the ratio and not the ratio. Please correct it.

-       The author should explain the differences in aSMase and nSMase activity in HREC and BV2 cells.

Author Response

Comments and Suggestions for Authors

In this paper Mondal and collaborators investigated differences in sphingolipids metabolism between human primary retinal endothelial (HREC) and murine microglial cells (BV2) in normal conditions and under high glucose-induced stress. The manuscript is generally well written and the sphingolipidomics data are interesting and promising. However, at this stage the paper is just descriptive and the results are just a picture of what happens in the two cell lines in the presence of high glucose. The main weakness is the lack of mechanistic insights. Furthermore, a very important weak point is the choice of comparing two cell lines of different origin (human and murine) which, according to the authors, may be at the basis of the differences observed in the SPL profile.Therefore, I have some concerns that should be clarify by the author:

The story depicted in the discussion for HREC and neovascularization when cultured in high glucose, is nice, but so far is just a speculation. It should be further validated with silencing experiments of B4GALT5 and B4GALT6, VEGFA and PECAM, to evaluate the pathway suggested. The same applies for the increase of aSMase activity and inflammation and its relationship with the breakdown of the BRB. At this stage is only a speculation, no data is shown that can support this hypothesis. A loss of function experiment (i.e. silencing) in the presence of HG should be performed.

We thank the reviewer for recognizing our sphingolipidomics data as interesting and promising. We agree that our current study is descriptive and focuses on the characterization of sphingolipid synthesis and the differences between cell types and culture conditions under high glucose stress. However, our study, for the first time, provides valuable information on sphingolipid metabolism and signaling in two distinct cell types of the retina, significantly improving our knowledge and understanding of SPL metabolism and signaling that has received very little focus in the past. Our future efforts will focus on signaling pathways based on the significant findings presented in this manuscript, including loss of function experiments. However, the studies concerning the relationship between loss of the BRB and aMSase activity or VEGFA or PECAM are beyond the scope of the current work. Furthermore, they require significant efforts as the loss of BRB in itself is complex involving several other cell types.

Also for BV2 the data are nice and promising, but still preliminary and only descriptive. Moreover,  the authors should speculate more about the different SPL content in the two cell lines investigated. As I said above, if they think that the differences found can be due to the different species origins, why they did not use a human microglial cell line?

We have addressed a similar concern raised by reviewer 1. While we agree with this reviewer that we must do a comparative study using a human microglia cell line (e.g., HMC3 - CRL-3304); however, this work is beyond the scope of current work to establish the SL composition across cell types under glucose stresses. We have incorporated a helpful suggestion by reviewer 1 that lipidomes segregate predominately according to organ- rather than species origin (See discussion, lines 458-468).

The images in Fig.4 are not clear. Phalloidin staining of the cytoskeleton to see the morphology of the cells must be performed, otherwise we cannot be sure that the cells are just folded back on themselves. Moreover, showing only one cell per timing is not representative of the situation. A larger field with more cells must be acquired and shown. A quantification of the fluorescence intensity/field must be also performed.

We have addressed a similar concern raised by reviewer 2. We routinely perform immunocytochemical studies of endothelial and microglial cells (PMID: 31690344, PMID: 31346222, PMID: 30054947, PMID: 29997321), and we are confident about the morphology of the cells. To reduce the bias in interpreting our existing data, we have captured images at lower magnification to include several cells per field, quantified the labeling, and revised the figure. Our data confirm our previous observation that LacCer increases under high glucose stress. We firmly believe and expect the reviewer to agree that our re-analysis has significantly reduced the selection bias or staining artifacts, if any. Please see the result section to reflect the modifications and the revised Figure 4.

Minor:

the second paragraph of introduction should be reorganized. Microglial cells appeared out of the blue (line 58-59) and were then introduced and described later (line 70-71).

Thank you. This has been updated.

A 10% reduction of SM in HG-treated HREC cells is biologically relevant? The authors should discuss this issue.

We included the discussion on SM reduction in HG-treated cells ( Please see lines 506-522).

Fig. 6: The y-axis label seems to me to be the fold change of the ratio and not the ratio. Please correct it.

We have corrected the y-axis to reflect the fold change of the ratio. Thank you.

The author should explain the differences in aSMase and nSMase activity in HREC and BV2 cells.

We have added discussion of aSMase and nSmase activity in HREC and BV2 Cells (Please see lines 554-562).

Round 2

Reviewer 2 Report

All points have been appropriately adressed by the authors

Author Response

We Thank the reviewer for accepting our revision.

Reviewer 3 Report

I have read the revised version of the paper and I think the authors have made an effort to improve the manuscript. Overall, I am satisfied by the authors response to my concerns. However, I have still some concerns about figure 4. In my opinion phalloidin staining would have greatly improved the quality of the figure. Anyway, if the authors are not able to do so they should at least indicate the number of Immunofluorescent experiments performed and the number of the fields/image acquired and analyzed for the quantification.

Moreover, in the “Data analysis” it should be reported that the analysis for significance was carried out on raw data.

Author Response

We thank the reviewer for accepting our changes. We are sure that their comment and guidance have significantly improved the quality and clarity of the manuscript. We, unfortunately, could not accommodate phalloidin staining at this time, but phalloidin and cell organelle markers co-staining are included in our follow-up studies. We appreciate the suggestion.

We have included in the manuscript that we have done at least three independent immunofluorescence experiments per condition/ treatment. We captured 5-6 independent fields per experiment to perform image analysis. Line: 169-171; 284-285

The “Data Analysis” section is updated as suggested. Line: 181-182